# Trends in Bone Metastasis Modeling

**DOI:** 10.3390/cancers12082315

**Published:** 2020-08-17

**Authors:** Roberta Laranga, Serena Duchi, Toni Ibrahim, Ania Naila Guerrieri, Davide Maria Donati, Enrico Lucarelli

**Affiliations:** 1Unit of Orthopaedic Pathology and Osteoarticular Tissue Regeneration, IRCCS Istituto Ortopedico Rizzoli, Via di Barbiano 1/10, 40136 Bologna, Italy; roberta.laranga@ior.it (R.L.); davidemaria.donati@ior.it (D.M.D.); enrico.lucarelli@ior.it (E.L.); 2BioFab3D@ACMD, St Vincent’s Hospital, Melbourne, VIC 3065, Australia; serena.duchi@unimelb.edu.au; 3Department of Surgery, St Vincent’s Hospital, University of Melbourne, Melbourne, VIC 3065, Australia; 4Osteoncology and Rare Tumors Center, Istituto Scientifico Romagnolo per lo Studio e la Cura dei Tumori (IRST) IRCCS, 47014 Meldola, Italy; toni.ibrahim@irst.emr.it; 5Rizzoli Laboratory Unit, Department of Biomedical and Neuromotor Sciences (DIBINEM), Alma Mater Studiorum University of Bologna, Via di Barbiano 1/10, 40136 Bologna, Italy; 63rd Orthopaedic and Traumatologic Clinic Prevalently Oncologic, IRCCS Istituto Ortopedico Rizzoli, Via Pupilli 1, 40136 Bologna, Italy

**Keywords:** bone metastasis, 3D models, ex vivo models

## Abstract

Bone is one of the most common sites for cancer metastasis. Bone tissue is composed by different kinds of cells that coexist in a coordinated balance. Due to the complexity of bone, it is impossible to capture the intricate interactions between cells under either physiological or pathological conditions. Hence, a variety of in vivo and in vitro approaches have been developed. Various models of tumor–bone diseases are routinely used to provide valuable information on the relationship between metastatic cancer cells and the bone tissue. Ideally, when modeling the metastasis of human cancers to bone, models would replicate the intra-tumor heterogeneity, as well as the genetic and phenotypic changes that occur with human cancers; such models would be scalable and reproducible to allow high-throughput investigation. Despite the continuous progress, there is still a lack of solid, amenable, and affordable models that are able to fully recapitulate the biological processes happening in vivo, permitting a correct interpretation of results. In the last decades, researchers have demonstrated that three-dimensional (3D) methods could be an innovative approach that lies between bi-dimensional (2D) models and animal models. Scientific evidence supports that the tumor microenvironment can be better reproduced in a 3D system than a 2D cell culture, and the 3D systems can be scaled up for drug screening in the same way as the 2D systems thanks to the current technologies developed. However, 3D models cannot completely recapitulate the inter- and intra-tumor heterogeneity found in patients. In contrast, ex vivo cultures of fragments of bone preserve key cell–cell and cell–matrix interactions and allow the study of bone cells in their natural 3D environment. Moreover, ex vivo bone organ cultures could be a better model to resemble the human pathogenic metastasis condition and useful tools to predict in vivo response to therapies. The aim of our review is to provide an overview of the current trends in bone metastasis modeling. By showing the existing in vitro and ex vivo systems, we aspire to contribute to broaden the knowledge on bone metastasis models and make these tools more appealing for further translational studies.

## 1. Introduction

Cancer harbors variable heterogeneity and plasticity, which are both difficult features to reproduce in experimental models but essential for maintaining intra- and inter-patients’ differences and for accurately analyzing molecular mechanisms of the neoplastic transformation or for drug screening purposes. While substantial progress has been made in developing anti-cancer treatment strategies for primary tumors, such as breast and prostate cancer, the therapeutic interventions for metastatic cancer still present challenges. These challenges are primarily due to the tumor heterogeneity between and within individual patients, which results in significant differences in the tumor growth rate, invasive ability, drug sensitivity, and prognosis among individual patients [1]. For these reasons, there is a clear necessity to develop accurate cancer models to better investigate molecular mechanisms that lead to cancer onset/progression and to find more effective treatment strategies.

The growth of metastasis at secondary sites is the major cause of mortality in cancer patients, and it may be clinically evident even after years from the first diagnosis and therapy. Beyond lung and liver, bone is the third tissue most frequently affected by metastasis [2,3], thus impacting patients’ quality of life. In fact, several human cancers, such as breast cancer (BC), prostate cancer (PC) and lung cancer (LC), often metastasize to bone in their advanced stages [4]. These unfortunate events are related to considerable morbidity, such as pathological fracture, spinal cord compression, bone pain, and hypercalcemia, which, also due to the lack of curative treatment options, contribute to a worse outcome [4].

Unfortunately, the process by which cancer cells from the primary tumor acquire an osteotropic phenotype is still not well understood. Recently, Salamanna and colleagues reviewed different studies that attempted to determine some of the factors that participate in both cellular and molecular mechanisms of bone metastasis onset [5]. The ability to simplify the complexity and simultaneously retain the major pathophysiological features of metastasis is necessary to identify the critical factors in the acquisition of cancer metastatic potential. Recently, three-dimensional (3D) models and cancer organoids have been increasingly used as a simplified and reliable in vitro model system to study bone metastasis [5,6,7]. However, there are several limitations to the application of these models, such as the lack of a purely native microenvironment and of a well-defined extracellular matrix (ECM) [8]. There is no better model to accurately represent the original tumor than an ex vivo model, consisting of a specimen directly derived from patients and enriched with their own microenvironment components. The main problem when producing this model, beside the availability of the biopsies, is the difficulty of maintaining samples alive and proliferating while studying the characteristics of bone metastasis and its response to therapy.

Our review aims to provide (1) a general background on the bone metastatic process; (2) a comprehensive summary of the current 3D models and technologies showing their limitations and advantages; and (3) an insightful overview of the current ex vivo bone metastasis models. Taking into account the recent methods used in bone metastasis research, we want to underline the need for new systems that are more innovative, easier to handle, and could better recapitulate the cancer biology and bone metastasis. The primary goal of these models would be to increase the predictability of preclinical data by improving the characterization of the models, implementing new methodologies and integrating complementary models (e.g., ex vivo 2D and 3D assays) in drug development pipelines.

## 2. Metastasis Process and Bone Metastatic Microenvironment

The ability of cancer cells to leave a primary tumor, to disseminate through the body, and to seed new secondary tumors is universally recognized to be the basis for metastasis formation. Various and sometimes conflicting hypotheses have been proposed to explain different aspects of this process, but no single concept unravels the mechanism of metastasis in its completeness.

A pioneer study conducted by Stephen Paget in 1889 hypothesized that metastasis is based on the interplay between the so-called ‘seeds’ (namely the cancer cells) and the ‘soil’ (or the host microenvironment) [9]. Paget’s theory was subsequently challenged by others studies, such as Ewing’s and Isaiah Fidler’s research [10,11]. Other important findings revealed that primary tumors themselves are responsible for the formation of suitable microenvironmental conditions in distant sites, being determinant for the sustainment of cancer cells survival and proliferation before the establishment of the new colony [12,13,14]. Nowadays, this particular microenvironment is usually referred to as ‘pre-metastatic niche’ [13,15]. The metastatic niche theory suggests that a properly favorable microenvironment (pre-metastatic niche) supports tumor cells to engraft (metastatic niche) and proliferate at secondary sites (micro- to macrometastatic transition). In particular, the pre-metastatic niche results from the synergic interaction between the endogenous organ microenvironment and specific factors secreted by primary tumors [15,16].

The bone niche is populated by different kind of cells including stem cells, progenitor cells, mature immune cells, and supporting stromal cells [17,18,19]. To date, two primary niches have been described, namely the osteoblastic niche and the perivascular one; they are characterized by two diverse types of adult stem cells and their progeny: hematopoietic stem cells (HSCs) and mesenchymal stem cells (MSCs) [19,20,21].

HSCs are multipotent progenitor cells that can be found in adult bone marrow, peripheral blood, and umbilical cord blood. The hierarchical lineages of HSCs consist of myeloid cells, B lymphocytes, and osteoclasts [22]. The MSCs are multipotent cells that are able to differentiate into the mesenchymal lineage cells, which include osteoblasts, adipocytes, chondrocytes, fibroblasts, and other stromal cells [19,20]. Both cells’ lineages are connected to each other in the bone niche and work together to maintain bone homeostasis, sustaining in particular the osteogenesis, osteoclastogenesis, and hematopoiesis processes.

Bone is a hierarchically organized connective tissue; it contains four types of cells—osteogenic cells, osteoblasts, osteocytes, and osteoclasts—embedded in a matrix of collagen fibers and hydroxyapatite, as an inorganic component. Osteogenic cells differentiate into osteoblasts [23]. When included into the calcified matrix, osteoblasts undergo their terminal differentiation into osteocytes, changing their structure and function. On the other hand, osteoclasts are large multinucleated cells derived from the hematopoietic lineage (monocytes). Both osteoblasts and osteoclasts participate in the maintenance of bone physiologic homeostasis; in fact, bone tissue is continuously remodeled in order to maintain structure and calcium equilibrium, by osteoclast-mediated bone resorption and osteoblast-mediated bone deposition [24].

In case of cancer progression, this equilibrium is usually altered, leading to osteoblastic, osteolytic, or mixed metastatic lesions depending on the cancer origin and type [24]. In osteoblastic metastasis, commonly found in PC patients, the metastatic bone is characterized by the deposition of new tissue not preceded by bone resorption, resulting in excessive and disorganized bone formation [24]. Instead, osteolytic metastasis is mainly diagnosed in breast, lung, and renal cancers, and it is usually present uncontrolled osteoclast activity [23]. In most of the cases, the two processes coexist; thus, is not possible to classify bone metastasis as a single defined process, with the clinical prevalence of one over the other [25].

The development of malignant bone metastasis is recognized as a dynamic multistep process, in which a subpopulation of cancer cells from the primary tumor gain the capability to invade surrounding tissues, intravasate, survive in the bloodstream, and extravasate, giving rise to the metastatic colonization in a distant bone microenvironment [26]. Considering its complexity, a successful approach to study the intrinsic biology of bone metastasis is to separate the various steps of the cascade and to deeply characterize the bone metastatic microenvironment. Quiao and Tang extended the concept of “fertile soil“, proposed by Paget, to include three distinct microenvironments: the primary tumor microenvironment (PTM), the circulation microenvironment (CM), and the bone microenvironment (BM), each one distinguished by several key points that will be further discussed [27].

In the context of the PTM, the formation of metastasis involves a subgroup of osteotropic cancer cells with increased proliferative and migratory capacities [27]. Angiogenesis and epithelial to mesenchymal transition (EMT) are critically important in this phase. The first responds to the increased metabolic demand of cancer cells supporting local invasion and distant dissemination [28]. In contrast, the EMT consists of the cellular transformation from an epithelial phenotype with apical–basal polarization to a mesenchymal one characterized by high motility features [29]. The activity of cancer cells in the CM begins with intravasation and ends with extravasation. Tumor cells that trespass the normal vascular endothelium using the newly formed microcapillaries, become circulating tumor cells (CTCs), and invade the CM. Due to the overexpression of various surface receptors involved in pro-survival pathways [30], these CTCs are able to evade anoikis and survive in the CM. Once CTCs enter the BM, they are redefined as disseminated tumor cells (DTCs); here, DTCs can remain in a dormant state for several years [31]. This condition, also known as “dormancy”, can revert in case of stress circumstances, compromised immune system, and/or the activation of specific molecular pathways, leading to the formation of macrometastases. When in the osseous tissue, CTCs influence the pre-metastatic niche already developed in order to create a compatible niche to support the metastatic growth (metastatic niche). This process has been largely studied particularly for BC and PC bone metastasis and reproduced in the up-to-date available 3D model of bone metastasis formation [32] (Figure 1).

### 2.1. 3D Models

Three-dimensional architecture is fundamental for tissue and organ formation and intracellular functions [33,34]. This is also true for neoplastic tissues, where cancerous cells spontaneously grow and proliferate in a 3D structure within the host. Thus, an ideal 3D culture model should not only resemble oncogenesis and in vivo tumor cell growth, but also imitate the interactions between cells connected to the ECM. To date, several 3D strategies have been published, aiming to recapitulate as best as possible the in vivo environment and also to bridge the gap between standard bi-dimensional (2D) tissue culture and animal models [35,36,37]. All these studies highlighted the multiple differences between 2D and 3D systems; thus, the interest in developing always more realistic models is constantly increasing [33,38,39]. The use of 3D culture to recount the tumor bone microenvironment offers a deeper understanding of bone metastasis and cancer biology by more faultless modeling of dynamic cell–cell and cell–ECM interactions. However, there is still a lot of work to be done in 3D models research, specifically in finding more adequate biomaterials for models in terms of cellular growth, reproducibility, scalability, and cost [40].

To date, several approaches have been explored. These methods can be divided into different categories according to the technology used (Table 1; Figure 2). Generally, methods for 3D cell culture can be classified as either scaffold-free or scaffold-based, with the scaffold being constituted by organic or synthetic materials. 

The models can also be divided on the basis of whether they are on static or dynamic cultures. Static 3D cultures include the seeding of cells in a spheroid-like structure without the ECM and the seeding of cells in matrices or scaffolds. Dynamic 3D cultures include either cell free scaffold or scaffold-embedded cultured in bioreactors, or perfused in microfluidic devices (Table 1; Figure 2).

### 2.2. 3D Models and Applications to Bone Cancer Metastasis Studies

#### 2.2.1. 3D Scaffold-Free Systems-Cells Spheroids

The multicellular tumor spheroid (MCTS), developed by Sutherland and collaborators in the 1970s [90], has become a classic model system for cancer research and is the representative of the scaffold-free systems subgroup. Spheroids are aggregates of cells that grow in suspension and resemble a physiological neoplastic tissue organization with the outer zone containing proliferating cells, the inner parts populated by few dividing cells with lack of nutrients and oxygen, and a central necrotic area [91].

Different methods can be used to obtain spheroids from a vast range of cell lines [92,93]. Such spheroids are suitable for basic studies of physiology and metabolism, tumor biology, toxicology, cellular organization, and the development of bio-artificial tissues. They represent a valuable tool to mimic the tumor microenvironment while incorporating different types of cells, having a high reproducibly rate and low related costs. However, not all cell lines form spheroids; some arrange in irregular cell aggregates. Moreover, spheroids can be considered too simplistic and not appropriate to simulate all the cell–cell and cell–ECM interactions that can affect the efficacy of anti-cancer drugs [94]. Moving the model to a fully embedded system offers further opportunities to mimic a tumor environment facilitating physiological cellular interactions, metabolism, growth, and metastatic invasion. There are multiple approaches to embedding spheroids, the most common based on the use of agarose, Matrigel, fibrin, synthetic polymers, and collagen [95]. The embedded spheroid model can be used to investigate the mechanisms of cancer cell invasion into the ECM. For example, they were used to monitor the invasion of glioma cells into collagen type I via imaging techniques [41]. There have been some studies showing 3D spheroids in combination with 2D endothelial cells that are able to form tubule-like structures that mimic vessel sprouting and angiogenesis [42]; however, there are no particular examples of this with bone metastasis models.

#### 2.2.2. Organoids

An organoid can be defined as a 3D-structured group of cells derived from primary tissue, embryonic stem cells (ESCs), or pluripotent stem cells (PSCs), with self-renewal and self-organization abilities, which are provided with similar functionality of the organ of origin. Organoids are innovative 3D in vitro culture platform and can be important tools for modeling disease such as cancer. They can recapitulate tumor heterogeneity and microenvironment, overcoming established cell lines, and do not require the excessive time and costs of in vivo patient-derived tumor xenografts (PDTXs). Furthermore, they can be stocked as “living biobanks”.

Since the discovery of colorectal cancer (CRC) organoids [96], 56 tumor-derived organoids have been established (including prostate, bladder, ureter, kidney, colon/rectum, brain, pancreas, breast, stomach, esophagus, soft tissue) [97,98]. Nevertheless, the majority of published structures referred as organoids do not comprehend all cell types present in vivo, such as mesenchymal, stromal, immune, and neural cells.

Organoids mimic some, but not all, of the structures and functions of real organs and cannot recapitulate all the stages of cancer [99]. Firstly, they lack vasculature, which is essential to nutrient and waste transport. Secondly, not all cell types found in vivo can generate organoids. Thirdly, some of them replicate only the early stages of organ development. Taking into account all these limits underlines that organoids are not completely suitable for modeling bone cancer metastasis or cancer metastasis in general. However, patient-derived organoids (PDOs) may be a valuable tool to develop personalized treatments. In fact, cultured PDOs deriving both from disease-site biopsies or healthy tissue provide sufficient material for HTS application, or for in-depth phenotypic profiling allowing the identification of crucial mutations, therefore guiding therapeutic choices. Furthermore, the possibility of growing matched normal and diseased organoids from patients permit the screening also of combinations of different drugs that specifically target the diseased tissue, thus reducing side effects.

#### 2.2.3. 3D Scaffold or Hydrogels Based Tumor Models

Scaffolds are 3D structures that can be made of diverse kind of materials with tunable porosities, permeability, surface chemistry, and mechanical properties that are designed to mimic the microenvironment of specific tissues. They can be composed of both biological or synthetic materials. Biological scaffolds generally use naturally derived ECM (e.g., Matrigel and collagen) to help cell adhesion and 3D structures formation. In respect of synthetic devices, natural ones offer a more similar microenvironment to physiologic conditions in terms of growth factors, cytokines, hormones, and other molecules [98]. Meanwhile, synthetic scaffolds have several clear advantages over other natural gels for 3D cultures. Firstly, they generally provide higher reproducibility compared to biological ones. Secondly, synthetic materials allow superior control over the scaffolds’ biochemical and mechanical properties in 3D cell cultures. Thirdly, they possess high water content, permitting the transport of oxygen, nutrients, waste, and soluble factors, all of which are important to cell functions [98]. Scaffolds can be made through a variety of techniques (e.g., 3D printing, particulate leaching, electrospinning, casting), and by way of their tunable characteristics and material properties, they can influence cell adhesion, proliferation, activation, and differentiation.

Recently, several 3D matrix-assisted assembly models of bone metastasis have been developed on both naturally and synthetically derived matrices, particularly for prostate and breast cancer cell metastasis studies. These models utilize scaffolds to support metastatic cancer cell growth and to recapitulate tumor microenvironment complexity [45].

An example, performed on a fibroin-like scaffold, has been done by Talukdar and colleagues; they demonstrated that the interaction of breast cancer cells (MDA-MB-231) with the BM [human osteoblasts-like cells (MG63) and mesenchymal stem cells (MSCs)] varies with spatial organization, the presence of osteogenic factors, and stromal cell type [47]. Another study by Cox et al. showed that 3D collagen glycosaminoglycan scaffolds support the growth and mineralization of 4T1 breast triple negative carcinoma cells and that mammary cells are capable of osteomimicry. These interactions contribute to the ability of cancer cells to preferentially colonize the BM [48].

Several studies have focused on creating a system modeling the BM of the BC metastatic niche. For example, Marlow’s group developed a 3D-collagen matrix seeded either with human primary MSCs or immortalized cell lines representing cell types found in human bone marrow [49]. Other authors used a similar method with alternative biomaterials (e.g., silk-based biomaterials) [100].

One of the main problem to solve while designing a scaffold is the adequate supply of nutrients in order to cultivate a viable tissue and maintain an appropriate level of oxygen. As a consequence of the insufficient oxygenation, a typical condition of the inside of the tissue-engineered construct is hypoxia. Moreover, this is also a common feature in solid tumors, including BC [101]. Low oxygen tension drives the dissemination of metastasis, and it is associated to worse prognosis [102,103,104]. In particular, BC mainly metastasizes to bone, which is characterized by a hypoxic microenvironment, despite the presence of a high vasculated system. Indeed, oxygen tension in the bone is lower than the values reported for other tissues [from <1–6% (approximately 7–43 mm Hg) [105] versus 2% and 9% (14–65 mm Hg)] [106]. However, it is still difficult to monitor the regional oxygen gradients in the bone, and more accurate models are needed to unravel this topic [104].

Liverani and colleagues presented a biomimetic 3D tumor model based on macroporous scaffolds, which resembles the hierarchically organized structure of extracellular collagen [107,108]. There are two different types of BC cell lines examined here by the authors: MCF-7 (estrogen-positive BC cells, linked to good prognosis), and MDA-MB-231 (triple negative and aggressive BC cells) [109]. Their discovery shows in this experimental scenario that cells are able to build a hypoxic core niche, with modified growth dynamics, and that more aggressive cancer cells are selected, similarly to what happens during the in vivo disease progression [108].

To evaluate the metastatic development of endometrial, prostate, and breast cancer, Sourla and colleagues [110] inoculated the KLE human endometrial cancer, MCF-7 and ZR-75 human breast cancer, and PC-3 human prostate cancer cells into a 3D type I collagen gel system, which was previously loaded with MG-63 osteoblast-like cells. Their work demonstrated that in contact with a 3D collagen gel system loaded with osteoblast-like cells, PC cells produce morphological evidence of blastic reaction and of local invasion. Another remarkable example is the study performed by Thakuri et al. on a 3D hybrid hydrogel system composed of collagen and alginate to examine the invasive capability of BC. The authors demonstrated that human mammary fibroblast cells facilitated the migration of BC cells out of spheroids and into the surrounding matrix [50].

To recreate microenviromental condition in vitro and increase understandings of PC metastasis, Sieh et al. developed a co-culture system with PC3 and LNCaP prostate cancer cells on a medical-grade polycaprolactone–tricalcium phosphate (mPCL-TCP) scaffold, finding increased expression levels of several known biomarkers associated with PC cells’ bone metastasis [51]. Furthermore, the same co-culture system can be an indirect 3D in vitro model to analyze paracrine interactions between PC cells LNCaP and human osteoblasts (hOBs). In fact, including LNCaP cells into polyethylene glycol (PEG) hydrogel with hOBs is possible to examine PC cells forming a multicellular mass similar to a vascular tumor. Furthermore, the 3D structure allowed the diffusion of factors secreted by hOBs and LNCaP cells that influenced each other (paracrine interplay) [51].

Focusing on bone mineral hydroxyapatite (HA), one of the main components of human bone ECM, through a 3D composite scaffold made by polylactide-co-glycolide (PLG) and HA particles, it was demonstrated that HA can directly regulate key steps of BC bone metastases [5,46]. Moreover, in the same study authors showed that interleukin-8 (IL-8) is pivotal in this phase, providing mammary epithelium cells with a more aggressive behavior [5,46]. Another 3D bone-mimicking system was set up by Jin et al. using three human BC cell lines with different metastatic potential, namely MDA-MB-231, MCF-7, and transfected MDA-MB-231. In this case, authors showed that decreasing HA particles’ size and concentration positively influenced BC cells adhesion and proliferation [111]. In parallel, the co-culture of MSCs and MDA-MB-231 in the same 3D system showed that the presence of MSCs cause an upregulation of metadherin expression, which is a recognized metastasis-associated gene in BC [46].

Polymeric scaffolds use synthetic hydrogels or other biocompatible polymers to generate the physical supports for 3D cultures. The most common hydrogels used for 3D culture include PEG, poly (vinyl alcohol) (PVA), and poly (2-hydroxyethyl methacrylate) (PHEMA) [98,112]. Hydrogels are employed for different purposes in the fields of 3D tumor models; for instance, they are widely used for modeling angiogenesis and recreating metastatic BM due to the relative ease of supplying and handling [112]. Synthetic hyaluronic acid hydrogels have been found to support the metastatic potential of PC cells [113]. PLGA and poly(ε-caprolactone) (PCL) scaffolds are also used to study microenvironment stimuli acting on tumor cells, which is similar to the inhibition of proliferation and invasion with cytotoxic drugs [114]. Additionally, these materials can be used to examine stem-like features and EMT process in comparison to a 2D system. For example, PC cells cultured in a chitosan–hyaluronic acid matrix changed their features, developing invasion-like attributes, proving that hyaluronic acid likely promotes the metastasis, EMT, and drug resistance of PC cells, which is followed by the activation of downstream signaling involved in cancer malignancy [27,115]. Therefore, both natural and synthetic scaffolds are useful tools for bone metastasis studies; however, it is important to evaluate the most suitable material, probably by combining the two together [27].

### 2.3. Cultivation and Biofabrication Systems

#### 2.3.1. Bioreactors

The cultivation of cells in a bioreactor has two main objectives: cell expansion and improved cells functionality. Bioreactors’ technology allows effective cell expansion by monitoring important parameters: substrate consumption/metabolite production, cell growth, pH, temperature, and gas supply. In dynamic bioreactor systems, the application of a fluid dynamics to the cellular microenvironment prevents the accumulation of secreted biomolecules and shear stress that can induce different forms of differentiation. Spinner-flasks and stirred-tank bioreactors are dynamic culture vessels employed to grow co-cultures of tumor cells [116], tumor cells in 3D as spheroids [117], or scaffold-based cultures [118].

To date, there are several studies performed in bioreactors investigating the relationship between bone and cancer cells. A bioreactor-based co-culture system has beneficial characteristics for the study of the first stages of metastatic dissemination. For example, it can be used to study BC cells’ spread to bone tissue, namely adhesion on endothelial cells and intravasation, the exit from the bloodstream, and distant metastasis establishment.

In this scenario, Dhuriati et al. generated a bioengineered murine derived osteoblastic tissue (OT) model with comparable features to physiologic bone tissue, emphasizing the ability of MDA-MB-231 BC cells to infiltrate and form colonies into the OT when co-cultured in a bioreactor [64]. Further studies by Krishnan et al. were interested in making an estimation of the ability of a 3D bone surrogate made of osteoblasts and osteoclasts co-cultured in a bioreactor, which was to be used as a predictive tool of the early stages of BC metastasis to bone. With this purpose, they implemented the previously mentioned system set up by Dhurjati et al. comparing different co-culture systems, using murine osteoblasts MC3T3-E1 and human metastatic BC cells MDA-MB-231, or the metastasis-suppressed line MDA-MB-231BRMS1, observing the degradation of OT by BC cells [63].

Another interesting application of a dynamic bioreactor has been set up by Paolillo and colleagues [119], who used a multi-compartmental modular bioreactor in order to follow the adhesion process that naturally occurs during metastasis when circulating cancer cells adhere and colonize target tissues.

The authors proceeded by combining a millifluidic technique with a scaffold-based system: a milli-scaled chamber (LiveFlow^®^ system) for the fluidic culture of scaffolds and membranes under low shear stress. In particular, human fibroblasts were grown on a 3D polystyrene scaffold, which was placed at the bottom of the chamber and maintained in a dynamic culture condition. Then, stem-like cancer cells derived from dissociated breast (MCF-7) or lung (A549) spheroids were added to the system, and the inhibitory effect of integrin antagonists on cell adhesion was tested. Through this system, the authors were able to set up a useful model to explore the initial steps of the metastasis process that is also potentially useful for further drug screening research [119].

With the aim of reproducing the interaction between bone stroma, PC cells C4-2, human osteosarcoma cell line MG63, and a wild-type immortalized human osteoblastic cell line HS27A, Sung et al. co-cultured the different cell populations in a rotating wall vessel (RWN) bioreactor system. The presence of C4-2 cells induced notable changes in both normal and osteosarcoma bone stromal fibroblasts, both in terms of phenotypic and molecular features. These results highlight how in 3D conditions, it is possible to reproduce the interplay between the different population of cells that modulate neoplastic growth and metastasis [65].

Typically, RWV bioreactors are employed to minimize physical forces; instead, another kind of bioreactors is able to recreate the intrinsic forces present in a tumor microenvironment. The importance of these interactions is underlined in direct co-culture studies where cancer cells and osteoblasts are in contact. Instead, several authors chose to divide the bioreactor in separate compartments through a cellulose membrane to observe the growth of OT alone or in co-culture with metastatic BC cells [90,113,120]. Therefore, bioreactors are confirmed as a useful method for studying the interaction between bone and metastatic cells.

#### 2.3.2. Microfluidic

Microfabrication techniques and microfluidic technologies have been combined to create microstructures that are able to manipulate small amounts (10^−9^ to 10^−18^ L) of fluids [78]. The microfabricated hollow channels can host different kind of cells and tissues, and they can be manufactured to control cell shape and function. Moreover, these platforms allow controlling dynamic fluid flows and spatiotemporal gradients in a 3D culture microenvironment [77].

There are several applications for cancer studies [121]; in particular, some of the most important steps of cancer metastasis can be simulated through these innovative systems. The continuous perfusion of media by the microfluidic network [122] can mimic blood flow, resulting in a higher ability of cells to extravasate [123,124]. Moreover, these devices can support 3D aggregates growth and the co-culture of multiple cell lines in the same chip [94]. An indicative example of microfluidic application is the possibility to recreate tumor–endothelial cells interaction, which is fundamental for understanding the process of metastasis formation, including angiogenesis, intravasation, and cancer cell colonization [71]. Furthermore, microfluidic systems have also been studied in detail to better recapitulate the cancer cell–immune cell interactions, with the ultimate aim of increasing knowledge on cancer immunotherapies [125]. In general, these kinds of systems are usually modeled to contain a microvascular network used for organ-specific extravasation experiments and the co-culture of multiple cell lines in the same chip.

A remarkable example of their application is the co-culture of tumor cells with endothelial cells human umbilical vein endothelial cells (HUVEC) to study the intravasation process. Specifically, it has been shown that the endothelial cells are able to form a lumen-like structure, and the tumor cells in contact, under low fluidic conditions, can migrate toward the lumen [126].

Furthermore, microfluidic platforms can be modified in order to generate more complex and dynamic models. For example, Bersini et al. implemented a microfluidic system with three media channels and four gels to co-cultivate different types of cells. For this study, in particular, MSCs and human umbilical vein endothelial cells (HUVECs) were used [72]. As a validation of their system, the authors reported that molecular pathways critical for the extravasation of BC cells, mediated by cell surface receptor CXCR2 and bone-secreted chemokine CXCL5, were activated [72].

Subsequently, Jeon et al. improved the previously described system by adding a third cell line, namely the osteodifferentiated primary MSCs, and embedding the HUVECs in a fibrin gel in order to resemble a BM matrix [73]. In this way, the authors were able to recreate a human bone-like microenvironment in vitro. Additionally, the perfusion flow that was generated and maintained into the microfluidic circuit allowed the recreation of a molecular gradient, influencing both cancer and endothelial cells. Finally, the introduction of MDA-MB-231 BC cells allows observing the extravasation process.

#### 2.3.3. Organ-on-a Chip

The in vitro cultures combining organoids and microfluidics, broadly known as organ-on-a-chip (OOC) models, have been extensively used to study various characteristics associated with tumor progression such as growth, angiogenesis, migration, metastasis, and drug response. An OOC can be defined as a device containing both cells and ECM, recapitulating tissues and organs in their original structure [6,127]. This model has many advantages: more specifically, it allows a precise control of the microenvironment, continuous flow perfusion culture, and it is suitable for high-throughput applications. Additionally, it allows maintaining some of the main tumor microenvironment (TME) features present in vivo such as multicellular interactions, ECM-based biochemical properties (by using biomaterials to encapsulate the cells), biophysical signals and their gradients [128], hypoxia [129], and others [76].

Notably, OOCs can be fabricated as platforms made of different mini-organs in the multiple micro chambers linked via microfluidic channels to form a human microphysiological system. This provides an extraordinary platform to study cancer multiorgan metastasis. Nevertheless, at present, most of these systems are not set up for ex vivo specimens but still utilize primary cell lines or stem cell-derived cells; therefore, they cannot totally mimic the histological and cellular features of native organs and tumors [130].

Below, we report several examples of OOCs to study tumor multiorgan metastasis and cancer–microenvironment interactions. To determine the mechanism of multiorgan metastasis from BC, Jeon et al. presented a model where endothelial cells, MSCs, and osteoblast-differentiated cells (OBs) were cultured in 3D ECM to mimic bone marrow and muscle microenvironments with the microvascular networks. Extravasation rates of metastatic BC cells were investigated on these microenvironments with or without adenosine treatment. The authors reported that metastatic BC cells exhibited distinct extravasation rates in relation to different microenvironments. Moreover, the inhibition of the A3 adenosine receptor in BC cells resulted in an increased extravasation rate in the muscle microenvironment [73].

In another study, a four OOC system was developed to model the metastasis of primary LC to the downstream organs, including the brain, liver, and bone [79]. The results showed that metastasis occurred in all four organs and displayed spatiotemporal heterogeneity over the different locations. Nevertheless, all these examples are still made of cancer cell lines and could not represent the critical features of the native tumor. In turn, the incorporation of metastatic tumor organoids with other OOCs presents a better way for studying cancer multiorgan metastasis [131].

#### 2.3.4. Bioprinting

3D bioprinting has arisen as an innovative method to develop functionalized living tissues and organs. Recent works highlighted the potential of this platform to study bone disease, although a lack of reports on cancer bone metastasis is still present [132].

Different approaches can be followed using an alternative combination of bio-inks, consisting of cells and gel-like materials [89]. Bioprinting technology can be used to study angiogenesis and invasion mechanisms, as demonstrated by Zhang et al. [133] and Mou et al. [83] for MCTSs and LC cells, respectively. In addition, it has been shown that 3D bioprinting fibers embedded in hydrogels are suitable to generate microvessels, which is useful to study cancer angiogenesis [82]. Another study, utilizing BC cells cultured on a hydrogel substrate using a pre-custom bioprinting platform, proved the feasibility of obtaining uniform spheroids, representing a controllable and high-throughput approach for modeling the TME [84].

Another remarkable application of this technique is in the study of osteotropic cell migration during bone metastasis. Through the use of 3D bioprinting, Huang et al. created a matrix with similar features to the CM that was able to simulate the vascular pattern demonstrated in vitro [85]. The authors showed that the caliper of the bioprinted microvessel was determinant in regulating the speed of cancer cells. Another study, conducted on a bioprinted PEG scaffold, showed how invasion is also modulated by the composition of the scaffold (in particular, its stiffness) and the morphology of the cells [134].

Combining 3D bioprinting and biomaterials, Zhu et al. fabricated a series of in vitro bone matrices composed of PEG hydrogel and different concentration of nanocrystalline hydroxyapatite (nHA). These scaffolds were designed to best recapitulate the native BM for the investigation of BC bone metastasis. The researchers used a stereolithography-based 3D printer to fabricate a bone matrix with finely tuned architecture; this optimized matrix was used to analyze the interaction between BC cells MDA-MB-231 and human fetal osteoblasts (hFOB). BC cells co-cultured with hFOB cells on the matrix directly affected the morphology, proliferation rate, and cytokine secretion of osteoblasts. IL-8 secretion by osteoblasts was enhanced in the presence of MDA-MB-231 cells. Moreover, the authors reported that the cellular organization in a 3D matrix was different compared to the monolayer culture; in fact, BC cells co-cultured with osteoblasts within the 3D bone matrix formed multicellular spheroids [135]. These results demonstrated the reliability of this kind of 3D printed bone matrices to study bone metastasis evolution.

### 2.4. Ex Vivo Models

Ex vivo models consist of freshly isolated biopsies both from healthy and/or diseased tissues derived from patients. In these tumor sections, the heterogeneity of the original tissue and the components of the surrounding microenvironment are preserved. These features make the ex vivo cultures a potent tool for personalized therapies [136].

To date, several approaches to culture ex vivo models have been proposed, including tissue slices and explant cultures. Although these methods have been applied to different fields of research [137,138], there are still technical problems and limitations to take into account (Figure 3) [139].

In general, the ex vivo culture setting allows maintaining the interactions within tumor cells and the microenvironment. Unfortunately, the cells survive only for a few days [140]. In particular, it is known that maintaining biopsies in culture preserving their viability and integrity is very complex. One of the main challenges is to maintain an adequate level of oxygen within the tissue explant. Fluctuations in its concentration result in modified cell features; on the other hand, oxygen variations are typical of each tissue. In particular, oxygen levels are very low in bone tissue in comparison to other sites. Moreover, commonly, in vitro culture conditions are maintained under a determinate amount of oxygen (usually around 21%) [141], which is not representative of most tissues and especially not for bone. Indeed, this is a limit in a long-term monitoring of disease progression and in drug treatments. Furthermore, patient-derived biopsies are often difficult to achieve, and the isolation and culture methods need to be optimized for each tissue type [110]. Complex methods can lead to difficult interpretation of the results, mainly due to the intrinsic heterogeneity of the samples. [142]. The main advantages and limitations of the ex vivo models are reported in Figure 3.

#### 2.4.1. Main Application of Ex Vivo Explants: Viability for Long Term Culture, Analysis of Tissue Architecture, and Response to Therapies

To date, few examples of primary tumor ex vivo models are reported in the literature. These models are mainly used to investigate tissue architecture, viability for long-term culture, and response to therapies.

Usually, tissues are cultured in the form of 200 μm thick slices both patient- or patient-derived xenograft (PDX)-derived and can be cultivated directly in culture medium or maintained in an air–liquid interface [140,143]. In order to improve cell viability and tissue organization, tissue biopsies can be cultured in a stirred system (approximately 150 rpm), which prevents oxygen zonation [144]. Moreover, tissue slices can be maintained on an air–liquid interface, nesting on top of a filter [144], which acts as a physical support, reducing cell death and vacuolation [144,145].

To better study tumor tissue architecture, Tanos et al. skillfully demonstrated the ability to maintain BC explants in culture up to 7 days in low-adhesion plates [146]. In this study, the authors managed to keep apico–basal polarity and a microlayer of myoepithelial cells, which is typical of luminal epithelial cells architecture. Moreover, thanks to the presence of a breast-like tissue organization, it has been possible to preserve cell–cell communication and hormone responsiveness, which are mechanisms that were not observed for BC cell lines nor for dissociated breast tissue. In conclusion, this work reinforces the need to keep the original tissue architecture to address hormone action in the breast and to preserve the original features of the tissue [146]. In order to further improve these kinds of systems, perfusion-based bioreactors can be employed to extend the culture time of BC specimens up to 15 days [147]. In this system, tissue integrity and cell viability were preserved, although cellularity was decreased alongside time. In another study, colorectal and head and neck cancer explant cultures were proposed as a co-clinical tool for the prediction of patient-specific drug response [148]. In this model, explants were cultured for 72 h, conserving the original phenotype in a matrix-specific scaffold, which was adapted for each tumor type. As a means to satisfy the needs of cytokines and growth factors, tumor explants were cultured with the patient autologous serum. By combining the results obtained with a machine learning algorithm, the authors obtained a prediction tool to evaluate tumor response to a specific drug treatment.

#### 2.4.2. Ex vivo Bone Metastasis Models and Applications

Ex vivo bone organ cultures were firstly developed in the 1990s with the purpose of modeling the interactions between cancer cells and the bone niche [101,149]. These first models were used to study bone growth, bone and cartilage matrix turnover, and the effects of cancer in bone [150,151,152]. More recently, this system has been employed to study mechanical loading, interactions between different bone cell types, the molecular steps involved in bone resorption/formation, and cancer progression [137]. In the past decades, a variety of creative assays have been developed to efficaciously model an in situ bone environment. The use of primary bone tissue in such experiments offers a direct way of sampling the skeletal milieu and where the process of metastasis can be studied [153]. Ex vivo assays using fresh bone such as fetal rat long bones [154], mouse calvariae [149,155], or embryonic metatarsals [156], have been used successfully to investigate cell-specific and heterogeneous populations, to test potential bone anabolic agents, or to study vascular differentiation and growth [153].

Firstly, murine calvariae specimens dissected from postnatal mice have been employed to model the tumor–bone niche. A fresh bone tissue was co-cultured in close proximity, but not in contact, to cancer cells up to 7 days. This created a two-compartment static system that allows for paracrine signaling between the fresh bone and tumor cells [157]. Bellido and colleagues further developed this model, switching to a rotating culture system. This allowed a direct contact of freshly bisected postnatal calvariae and tumor cells, in order to stimulate bone–tumor cell interactions [137]. A recent work has extended this model to a more accurate system that is focused on the human bone–tumor niche, by replacing rodents’ samples with human ones [158]. Uniform discs derived from fresh human bone samples were co-inoculated with human PC cell lines, cultured over 7–14 days, and the distribution of the different kind of cells was observed under study. This model shows that the adherence of PC cells occurs both in trabecular bone regions, by forming independent tumor colonies similar to those observed in human metastatic disease, and through direct interaction with the bone matrix, by invasion of the cellular microenvironment and trabecular bone structure of the bone core slices [158].

The incorporation of species-specific microenvironments attracting secondary tumor invasion is another important step of the metastasis cascade under study. In an interesting work, Tulotta et al. developed a novel humanized mouse model of ER+ and ER– BC PDX that was able to grow at the primary site and spontaneously metastasize to a human bone environment [159]. In this model, the authors mixed different approaches: 3D ex vivo PDX, 2D in vitro BC cells, and ex vivo bone specimens. In particular, to obtain the human bone environment, bone discs from femoral heads obtained from patients subjected to hip replacement surgery were implanted subcutaneously into NOD/SCID mice. To study the metastatic process, 7 PDX tumors and 3 BC cell lines (MDA-MB-231-luc2, T47D-luc2, or MCF7-Luc2 cells) were injected into the fourth mammary duct; the development of metastases was followed by luciferase imaging and confirmed on histological sections. By analyzing bone integrity, viability, and vascularization, it was demonstrated that after 4 weeks, bone implants were alive, re-vascularized, and remodeled (as evidenced by the presence of osteoclasts, osteoblasts, and calcein uptake). Moreover, by analyzing the expression profile of genes and protein during different stages of metastasis, it was possible to demonstrate that BC cells underwent a series of molecular changes through the metastatic process; this finding identifies molecules as useful metastatic drivers that could be helpful in predicting future relapse in bone in BC patients. This highlights the potential impact of this humanized model system in delivering translatable data from the laboratory to the clinic [159]. Thanks to human bone xenotransplantation, the maintenance of an osteotropic phenotype was demonstrated to be a key strength of this model. Furthermore, Lefley and colleagues showed that cells not only metastasized to human bone implants but also formed metastases in mouse bone [160]. In conclusion, the authors demonstrated that the implantation of femoral bone provided a metabolically active, human-specific site for tumor cells to metastasize. Additionally, they proved that this novel model could provide significant advancements in the modeling of BC bone metastasis [160].

#### 2.4.3. A Comparison between In Vitro and Ex Vivo Models

As mentioned above, ex vivo bone metastasis cultures are a convenient alternative that bypasses the limitations of in vitro models, including 2D primary cell culture and 3D culture. In vitro models do not entirely reproduce the cellular diversity, and the complexity of interactions present in the tumor niche and tumor heterogeneity is still difficult to be represented ex situ in a 3D model. Nevertheless, 3D models can add more complexity and scalability when compared to 2D models. Additionally, compared to 2D, the tumor environment can be better reproduced in a 3D model; this is because in a 3D model, the tumor cells are isolated to grow heterotopically. Additionally, thanks to the current technologies developed, 3D models can be scaled up for drug screening as effectively as the 2D systems.

Compared to in vitro models, cultured bone explants could better mimic the original features observed in patients; in fact, they can maintain the natural position of osteocytes within the extracellular mineralized matrix, thus retaining the in vivo 3D distribution and the innate proportion of osteocytes compared with other cells (osteoblasts, osteoclasts, bone marrow cells, and endothelial cells). Furthermore, organ cultures using human bone reproduce human conditions and are a useful tool to test patients’ responses to therapeutic agents. Ex vivo bone organ cultures are an effective model to simultaneously examine the anti-cancer efficacy and bone effects of therapies before moving to in vivo experimentation. In comparison to animal models, ex vivo organ cultures faithfully reproduce results seen in vivo and represent much less expensive screening models. On the other hand, as previously described, working with patient-derived explants is not as easy as working with cell cultures, either in terms of sample availability or in terms of experimental analysis and result interpretation [142].

It is important to underline the potential of both cell culture settings to be included as preclinical testing models, alternatives, or adjuvant to animal models, to challenge the inter- and intra-tumor heterogeneity of biological behavior and treatment response in human cancer. Despite the enormous contribution of PDXs in improving the knowledge in cancer research, their use on a large scale for drug screenings is not possible, from an animal welfare point of view as well as for financial reasons. 3D models and ex vivo cultures derived from patients’ tissue retain genomic features and in vivo drug response and can systematically and consistently be generated. This animal-free and cost-effective drug development pipeline for HTS prior to in vivo testing using PDXs models is currently still under investigation across cancer types and open for improvement.

## 3. Discussion

In the last decades, the study of cancer biology has progressed from 2D culture on glass plates to complex 3D models of tissues, organs, and body systems. The numerous 3D models developed have allowed for a deeper knowledge of cancer biology.

Several inherent obstacles impede the study of cancer metastasis in bone. They include the physical difficulty of manipulating bone as a tissue, the complexity of obtaining *ante-mortem* samples of bone metastases from human patients, the limited number of representative models that effectively mimic human disease, and the inconvenience of identifying or monitoring cancer cells using in vivo or ex vivo models. Furthermore, bone metastasis studies are also limited by the amount of time that the model can be maintained in culture. Molecular oxygen is one of the most important variables to consider. In fact, oxygen transport is limited in vitro either by culture medium or by tissue thickness, leading to hypoxic areas, therefore restricting the viability of the explant. Several strategies can be followed to overcome these problems, such as cultivating the samples into a bioreactor, using specific scaffolds resembling the microvasculature or oxygen carriers [102]. Thus, it is difficult to achieve real-time monitoring using in vivo or ex vivo models. In addition to this, a main disadvantage to take into account is the scalability of the ex vivo models; considering the low availability of tissue samples from metastatic patients, high-throughput analysis is almost impossible, while the personalized medicine opportunity is relevant. For instance, with the ex vivo model of bone metastasis, it is possible to study the specific response of patients using their own avatar of the metastasis in the lab before attempting drug regimens or novel treatment.

A few in vitro models that can overcome the aforementioned shortcomings have been established for bone metastasis studies, and the existing in vitro models are unable to fully replicate all the pathological conditions. Simple 2D monolayer culture models failed to mimic the bone, as it is a complex elastic tissue where different populations of living cells are embedded in a dynamic environment. Most of the in vitro models currently used mainly include endothelial cells, not allowing a complete study of the bone microenvironment. Commonly used in vivo models of PC and BC metastasis include syngeneic rodent cancers and xenografts of human cancer in immunodeficient mice. Unfortunately, these models rarely develop bone metastasis, and in the case of xenografts, differential factors between species may inhibit the capacity of human cells to colonize murine bones.

Currently, the therapeutic approaches used to limit bone metastasis development aim to restore the balance between osteoblast and osteoclast cells populations, such as bisphosphonates and denosumab treatments. However, considering the presence of mixed metastatic lesions, the inhibition of the osteolytic process could not be sufficient [161].

Several models that are able to mimic bone metastasis are under investigation. As demonstrated by in vivo preclinical bone metastatic models of PC, it is possible to reproduce different aspects of the disease using a variety of established cell lines. In particular, some are useful to study osteolytic lesion, while others better resemble osteoblastic ones [162]. However, it is important to consider that human PC bone metastasis is mainly osteoblastic, so cell lines that induce osteoclastic lesions (for example, PC3 cells) cannot fully represent what happens in patients [162]. Additionally, cells after series passages can lose their heterogeneity, and the use of immunodeficient animals is challenging and cannot recapitulate host immune response. Therefore, 3D in vitro models are interesting alternatives to replace animals in PC bone metastasis research, as argued previously [110,116]. Regarding BC bone metastasis models, co-culture of cells in microfluidic platforms seems to be the elective method to describe osteoblastic lesions [72,73], while none of these models include osteoclastic cell lines [161,163]. A bioreactor-based co-culture system has also beneficial characteristics for the study of the first stages of metastatic dissemination [64,164]. Combining different co-culture systems made of osteoblasts and human metastatic BC cells, it is possible to observe the degradation of bioengineered osteoblastic tissue by BC cells [63]. In addition to these, an alternative approach may be the co-culture of bone tissue explants with cancer cells, preserving the natural microenvironment of bone. Furthermore, it is possible to engineer a two-compartmental model to follow the paracrine signaling between bone and cancer cells, and their osteolytic or osteoblastic differentiation [157,158,165]. Despite the aforementioned advantages, this method is not broadly used, probably because of the scarce availability of fresh bone tissue explants derived from patients, which are necessary either to maintain a bone-specific microenvironment or species-specific osteotropism [5].

As a consequence, there is not a particular model system that could be more suitable to reproduce osteoblastic rather than osteolytic lesions. In fact, to date, a modular approach is generally adopted in order to recapitulate different steps of the metastatic cascade using translational models [162]. For the reader’s convenience, a detailed description of these models of bone metastasis has been published by Berish et al. [162]. The ability to make this intricate mechanism easier while at the same time retain its major pathophysiological features is mandatory to identify the critical factors in the acquisition of cancer metastatic potential.

In this review, we have discussed the evolution of 3D models, with a specific focus on bone metastasis. We explored the main 3D models currently in use, going into their efficacy, limits, and potential. Lastly, we analyzed the ex vivo models of bone metastasis and we focused on the aspects that must be critically examined and improved, so that they can exert greater clinical impact in the future. In our opinion, together with emerging 3D in vitro platforms, the use of ex vivo models might help uncover variables inherent to tumor heterogeneity and metastasis, in order to corroborate hypotheses formulated from studies employing less complex in vitro models. Especially, ex vivo models could be used as avatar metastases for the prediction of drug response for personalized medicine.

In fact, the coordinated action of multiple cell types, growth factors, and ECM present in the native neoplastic BM might influence cancer cells’ behavior differently to what usually happens in standard in vitro models. In parallel, the analytical tools used for the interpretation of results need to be improved in order to simplify the analysis process.

## 4. Conclusions

Metastasis is a common consequence of many types of tumors. A major challenge in the clinical translation of potential anticancer drugs and treatments is the discrepancy in the in vitro to the in vivo efficacy of candidates. For this reason, there is an urgent need for stable and precise models recapitulating cancer onset and metastasis. To this end, 3D preclinical models have emerged as promising technologies attracting increasing multi-faceted recognition for its utilization in cancer biology and metastasis studies by allowing for the recapitulation of the in vivo TME at high reliability outside the human body. However, ex vivo patient-derived models are the only system that is able to fully resemble inter- and intra-patient heterogeneity. The motivation here is to improve our capacity in bone metastasis modeling over existing strategies, in attempts to facilitate drug development as well as personalized therapeutic screening, especially for the latter, where the use of ex vivo models may be conveniently combined with patient-derived cells, 3D dynamic systems, and materials. Indeed, ex vivo models are anticipated to bridge the gap between conventional 2D cell cultures and animals, and they can possibly even replace animal models (e.g., PDXs) in the case of personalized medicine.

## Figures and Tables

**Figure 1 cancers-12-02315-f001:**
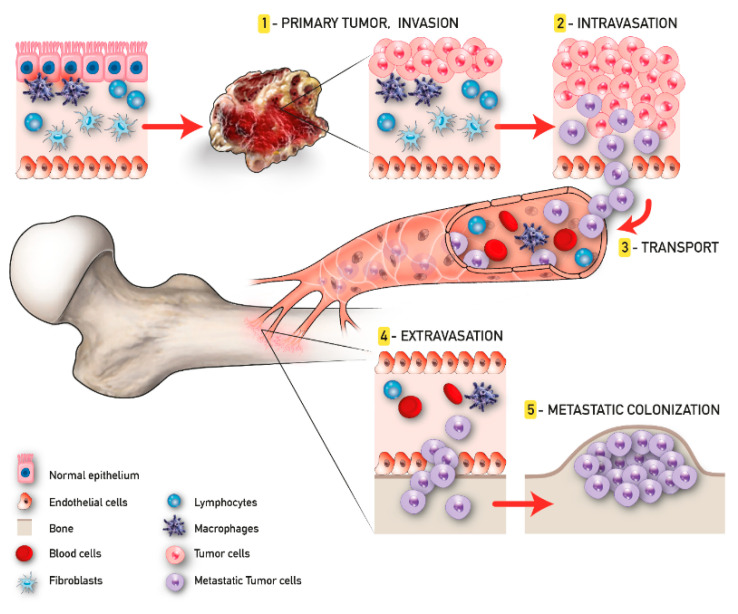
Tumor progression and bone metastasis process.

**Figure 2 cancers-12-02315-f002:**
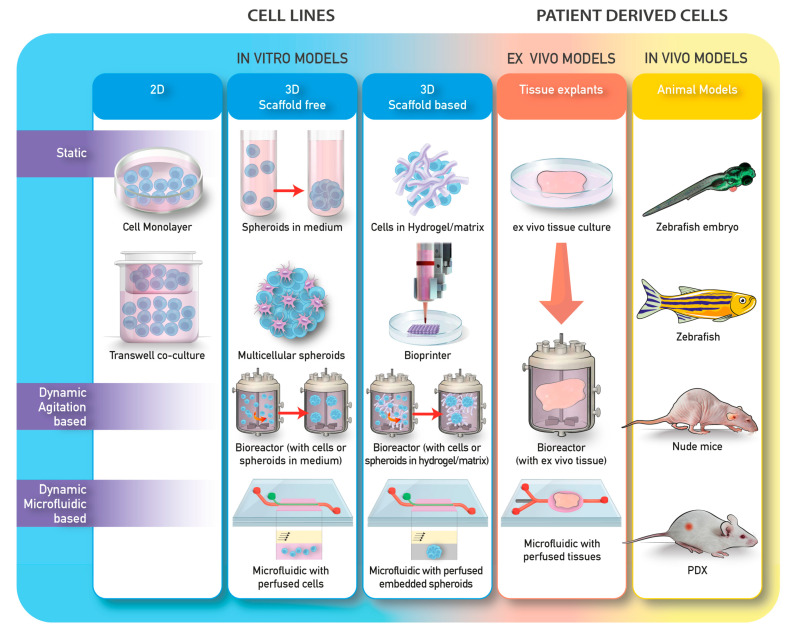
Examples of in vitro, ex vivo, and in vivo models commonly used in research.

**Figure 3 cancers-12-02315-f003:**
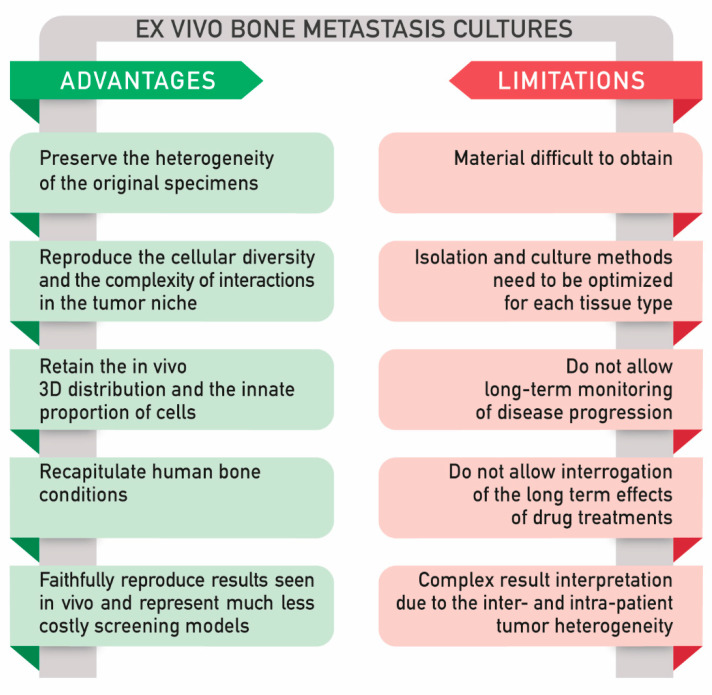
Ex vivo bone metastasis cultures, advantages, and limitations.

**Table 1 cancers-12-02315-t001:** 3D Model Technologies Advantages and Limits. BC: breast cancer, BM: bone microenvironment, ECM: extracellular matrix, HA: hydroxyapatite, IL: interleukin, OT: osteoblastic tissue, PC: prostate cancer, TME: tumor microenvironment.

3D Models	Cancer Feature Studied	Pros	Cons	Other References
**Scaffold** **free**	Cancer cell invasion into the ECM [41]; Formation of tubule-like structure that mimics vessel sprouting and angiogenesis [42]; Collective invasion and intravasation [43]; Collective invasion [44].	High Reproducibility; Low cost; Co-culture; HTS approach.	No support or porosity; Not optically transparent; Poor control over spheroids/organoids size; No cell/ECM interactions.	[45,46,47,48,49,50,51,52,53]
**Scaffold** **based**	Cell adhesion, proliferation, activation, and differentiation to hold up metastatic cancer cell growth and to resemble TME complexity [45]; Role of HA and IL8 in switching mammary tumor cells toward a more invasive phenotype [46]; Cancer cells and BM interplay is influenced by spatial organization, osteogenic factors, and stromal cell type [47]; Osteomimicry, the BM [48]; BM model of the BC metastatic niche [49]; PC cells in contact with osteoblast-like cells embedded in 3D collagen gel system produced morphological evidence of blastic reaction and of local invasion [50]; Evaluation of metastasis development from endometrial cancer, PC, and BC co-culture and expression of biomarkers associated with PC cells BM [51].	Co-cultures; Large variety of materials; Customizable; Affordable cost; High similarity to the in vivo conditions; Promotion of cellular attachment, proliferation, and differentiation; HTS approach sustainable.	Possible scaffold-to-scaffold variation; Not always optically transparent; Difficult cells removal; HTS options limited; Gelling mechanisms; Batch to batch variations; Undefined constituents in natural gels; Poor mechanical properties.	[52,53,54,55,56,57,58,59,60,61,62]
**Cultivation and Biofabrication Systems**	**Cancer Feature Studied**	**Pros**	**Cons**	**Other References**
**Bioreactors**	Reconstruction of a bone surrogate to study the early stages of BC invasion to bone [63]; Co-culture of OT with metastatic BC cells [64]; Reproduction of the interaction between bone stroma, PC cells, and human osteosarcoma cell line [65].	High similarity to the in vivo conditions; High volume of cells production; Customizable and controlled culture parameters.	Space required for dynamic cell culture; High costs for dynamic cultures; HTS options laborious.	[62,66,67,68,69,70]
**Microfluidic**	Angiogenesis, intravasation [71]; Study of molecular pathways implicated in BC cells extravasation, mediated by cell surface receptor CXCR2 and bone-secreted chemokine CXCL5 [72]; Microvascularized bone-mimicking microenvironment, defined by active differentiated bone cells, which generated spontaneously molecular gradients affecting both microvasculature and cancer cells [73]; 3D multicellular spheroid composed by PC-3 metastatic PC cells, osteoblasts, and endothelial cells [74].	Co-cultures (cell–cell, cell–tissue); Control of cell shape and function; Tune dynamic; Fluid flow and spatiotemporal gradient; Customizable; Commercial availability.	Required expertise; High cost of microfabrication; HTS options limited; Microenvironment parameters not measurable; Cell growing media for co-culture not well established.	[75,76,77,78]
**Organ-on-a-chip**	Tumor multiorgan metastasis and cancer microenvironment interaction [73]; Development of a four organ-on-a-chip system [79].	In vitro organ specific systems; High gas permeability; Optically transparent; Commercial availability.	Required expertise; High cost for the microfabrication; HTS options limited	[80,81]
**3D bioprinting**	3D bioprinting fibers embedded in hydrogels to recreate microvessels and study cancer-related angiogenesis [82]; Proliferation and invasion ability [83]; Modeling tumor microenvironment [84]; Migration of osteotropic cells during bone metastasis [85]; In vitro bone matrices to mimic the native BM for the investigation of BC bone metastasis [86].	Automated robotic processes; Spatially assembling multiple types of cells; Large variety of biomaterials and printing technologies; Bimolecular gradient production; Printable, crosslinkable, biocompatible and bioactive bioinks.	High cost; Required expertise.	[56,75,87,88,89]

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
