# Peer review of "Trends in Bone Metastasis Modeling"

_cancers, 2020, doi:10.3390/cancers12082315_

Round 1
Reviewer 1 Report
This is a well-organized and comprehensive review regarding the 3D model of bone metastasis. The manuscript described several experimental approaches and devices used to address the complexity of bone metastasis. Although many models mainly use tumor cells in their system, some models have included different cells and a co-culture system to investigate the interaction among cells in the metastatic environment. As we know that the biologic features of bone metastasis, such as osteoblastic vs osteoclastic bone metastasis, are quite different among various cancers, it is suggested to include discussion on whether particular model system is more adequate to address a particular bone metastasis feature or the limitation of different model system on these issues.
Author Response
We thank Reviewer 1 for the comments; we decided to address the subject adding two paragraphs, one in the introduction and the other in the discussion section.
Please find below the specific reference to our revision of the original text:
line 105: "Bone is a hierarchically organized connective tissue; it contains four types of cells, such as osteogenic cells, osteoblasts, osteocytes and osteoclasts, embedded in a matrix of collagen fibers and hydroxyapatite, as inorganic component. Osteogenic cells differentiate into osteoblasts [23]. When included into the calcified matrix, osteoblasts undergo their terminal differentiation into osteocytes, changing their structure and function. On the other hand, osteoclasts are large multinucleated cells derived from the hematopoietic lineage (monocytes). Both osteoblasts and osteoclasts participate in the maintenance of bone physiologic homeostasis; in fact, bone tissue is continuously remodeled in order to maintain structure and calcium equilibrium, by osteoclast-mediated bone resorption and osteoblast-mediated bone deposition [24].
In case of cancer progression, this equilibrium is usually altered, leading to either osteoblastic, osteolytic or mixed metastatic lesions depending on the cancer origin and type [24]. In osteoblastic metastasis, commonly found in PC patients, the metastatic bone is characterized by deposition of new tissue not preceded by bone resorption, resulting in excessive and disorganized bone formation [24]. Osteolytic metastasis, instead, are mainly diagnosed in breast, lung, and renal cancers, and usually present uncontrolled osteoclast activity [23]. In most of the cases the two processes coexist, thus is not possible to classify bone metastasis as a single defined process, with the clinical prevalence of one over the other [25]."
line 596:"Currently therapeutic approaches used to limit bone metastasis development aim to restore the balance between osteoblast and osteoclast cells populations, like bisphosphonates and denosumab treatments. However, considering the presence of mixed metastatic lesions, the inhibition of the osteolytic process could not be sufficient [161].
Several models able to mimic bone metastasis are under investigation. As demonstrated by in vivo preclinical bone metastatic models of PC it is possible to reproduce different aspects of the disease using a variety of established cell lines. In particular, some are useful to study osteolytic lesion while others better resemble osteoblastic ones [162]. However, it is important to consider that human PC bone metastasis are mainly osteoblastic, so cell lines that induce osteoclastic lesions (ex. PC3 cells) can’t fully represent what happens in patients [162]. Additionally, cells after serial passages can lose their heterogeneity and the use of immunodeficient animals is challenging and can't recapitulate host immune response. Therefore, 3D in vitro models are interesting alternatives to replace animals in prostate cancer bone metastasis research, as argument prevously [110,116]. Regarding BC bone metastasis models, co-culture of cells in microfluidic platforms seems to be the elective method to describe osteoblastic lesions [72,73], while none of these models include osteoclastic cell lines [161,163]. A bioreactor-based co-culture system has also beneficial characteristics for the study of the first stages of metastatic dissemination [64,164]. Combining different co-culture system made of osteoblast and human metastatic BC cells it is possible to observe the degradation of bioengineered osteoblastic tissue by BC cells. In addition to these, an alternative approach may be the co-culture of bone tissue explants with cancer cells, preserving the natural microenvironment of bone. Furthermore, it is possible to engineer a two-compartmental model to follow the paracrine signalling between bone and cancer cells, and their osteolytic or osteoblastic differentiation [157,158,164]. Despite the aforementioned advantages this method is not broadly used, probably because of the scarce availability of fresh bone tissue explants derived from patients, necessary either to maintain bone specific microenvironment or species-specific osteotropism [5].
As a consequence, there is not a particular model system that could be more suitable to reproduce osteoblastic rather than osteolytic lesions. In fact, to date, a modular approach is generally adopted in order to recapitulate different steps of the metastatic cascade using translational models [162]. For the reader convenience, a detailed description of these models of bone metastasis has been published by Berish et al.[162]. The ability to make easier this intricate mechanism but at the same time retaining its major pathophysiological features is mandatory to identify the critical factors in the acquisition of cancer metastatic potential."
In addition, we sent the manuscript to a professional for the english language and style revision.
Reviewer 2 Report
The atricle is well founded and easily readable for a broad audience.
I think the manuscript can be accepted for publication in Cancers, I only suggest to authors to better discuss the use of millifluidic systems and tranlational model of bone metastasis. These issues are likely to earn increasing importance in the future (eg:Stem-Like Cancer Cells in a Dynamic 3D Culture System: A Model to Study Metastatic Cell Adhesion and Anti-Cancer Drugs.Paolillo M, Colombo R, Serra M, Belvisi L, Papetti A, Ciusani E, Comincini S, Schinelli S. Cells. 2019 Nov 13;8(11):1434. doi: 10.3390/cells8111434.
Translational models of prostate cancer bone metastasis. Berish RB, Ali AN, Telmer PG, Ronald JA, Leong HS. Nat Rev Urol. 2018 Jul;15(7):403-421. doi: 10.1038/s41585-018-0020-2.)
Author Response
All the authors wish to thank Reviewer 2 for all comments and suggestions.
In order to better discuss the use of millifluidic systems we added the following paragraph in the "Bioreactors" subchapter:
line 328:"Another interesting application of dynamic bioreactor has been set-up by Paolillo and colleagues [119], who used a multi-compartmental modular bioreactor in order to follow the adhesion process that naturally occurs during metastasis, when circulating cancer cells adhere and colonize target tissues.
The authors proceeded by combining a millifluidic technique with a scaffold based system: a milli-scaled chamber (LiveFlow® system) for fluidic culture of scaffolds and membranes under low shear stress. In particular, human fibroblasts were grown on a 3D polystyrene scaffold, which was placed at the bottom of the chamber, and maintained in a dynamic culture condition. Then, stem-like cancer cells derived from dissociated breast (MCF-7) or lung (A549) spheroids were added to the system, and the inhibitory effect of integrin antagonists on cell adhesion was tested. Through this system the authors were able to set up a useful model to explore the initial steps of the metastasis process that is also potentially useful for further drug screening research [119]."
Regarding the discussion translational model of bone metastasis, we added a comment in the discussion section at line 622, refering the readers to the review article you suggested. A more detailed description of these models seemed to us out of scope for our work.
Reviewer 3 Report
The authors provide a very extensive and thoughtful review of current in vitro, in vivo and ex vivo techniques used in modeling bone metastasis, a frequent occurrence in multiple notable cancers, including those of the breast, prostate and lung. Throughout the manuscript, the authors present some examples of usage for the various models discussed, and provide insightful comparisons between each model, including limitations (e.g., cost, feasibility, etc). Overall, the figures are exceptionally well done; the Table could be revised a bit to make them easier to read (see below). I have a few recommendations that I believe will improve the quality and the impact of the manuscript:
- In general, the manuscript needs minor revision for vocabulary and grammar, and should be edited prior to resubmission.
- Figure 1: In the narrative, the authors describe the Seed:Soil hypothesis in some depth, but don’t describe the constituency of the microenvironment of the bone metastatic niche in the text. It might be helpful to briefly mention the different cell types of this particular tumor microenvironment (as partially depicted in Fig 1), as well as those cell types found in bone in general (osteoblasts, etc). This is especially important considering that most of the in vitro models described only involve endothelial cells, which is a limitation of these studies and support their argument that the path forward may best be through ex vivo
- Table 1: Please center the “Pros” column for Scaffold Free. Also, with the current alignment it’s a bit difficult to read regarding which feature has which pros/cons.
- Line 221: ECM abbreviation is never defined as extracellular matrix.
- The authors passively mention hypoxia in reference to BC in section 2.2.3.; bone marrow has an objectively substantial hypoxic environment of ~1-2% O2 in vivo [although it has also been reported as high as ~10%]. In most cases, in vitro culture occurs in ~21% O2, which is not representative of most tissues but especially not of bone. This is a significant limitation of current approaches, including in ex vivo studies, and should be thoughtfully addressed [with appropriate references] perhaps in the Discussion.
Author Response
The authors wish to thank Reviewer 3 for all the positive comments on our work.
Please find below the answers for each specific comment:
1- The manuscript has been revised for english language by a professional;
2- Thanks for the comment, we agree that a better explanation about the different population of cells that inhabit both the tumor microenvironment and the bone tissue would be helpful for the readers. We made the following additions to the original text in the introduction and in the discussion section:
line 95: "The bone niche is populated by different kind of cells including: stem cells, progenitor cells, mature immune cells, and supporting stromal cells [17–19]. To date, two primary niches have been described, namely the osteoblastic niche and the perivascular one; they are characterized by two diverse types of adult stem cells and their progeny: hematopoietic stem cells (HSCs) and mesenchymal stem cells (MSCs) [19–21].
HSCs are multipotent progenitor cells that can be found in adult bone marrow, peripheral blood, and umbilical cord blood. The hierarchical lineages of HSCs consist in myeloid cells, B lymphocytes and osteoclasts [22]. The MSCs are multipotent cells able to differentiate into the mesenchymal lineage cells, which include osteoblasts, adipocytes, chondrocytes, fibroblasts, and other stromal cells [19,20]. Both cells lineages are connected to each other in the bone niche and work together to maintain bone homeostasis, sustaining in particular the osteogenesis, osteoclastogenesis and hematopoiesis processes."
line 589: "Simple 2D monolayer culture models failed to mimic the bone, as it is a complex elastic tissue where different populations of living cells are embedded in a dynamic environment. Most of the in vitro models currently used mainly include endothelial cells, not allowing a complete study of the bone microenvironment."
3- We agree with the reviewer's suggestion, Table 1 has been restyled in order to be clearer and helpful for the readers;
4- The acronym ECM was already defined at line 66, but to be sure we check all the acronyms and definitions through the manuscript;
5- This is an interesting point, we are grateful for highlighting it. We decided to address the argument by explaining more in depth the involment of hypoxia in bone metastasis and the difficulty to reproduce it in in vitro and ex vivo models both in the introduction and in the discussion sections:
line 248: "One of the main problem to solve while designing a scaffold is the adequate supply of nutrients in order to cultivate a viable tissue and maintain an appropriate level of oxygen. As a consequence of the insufficient oxygenation, a typical condition of the inside of the tissue-engineered construct is hypoxia. Moreover, this is also a common feature in solid tumors, including BC [101]. Low oxygen tension drives the dissemination of metastasis and it is associated to worse prognosis [102–104]. In particular, BC mainly metastasize to bone, which is characterized by a hypoxic microenvironment, despite the presence of a high vasculated system. Indeed, oxygen tension in the bone is lower than the values reported for other tissues [from < 1% – 6% (~7 mm Hg – 43 mm Hg ) [105] versus 2% and 9% (14 – 65 mm Hg)] [106]. However, it is still difficult to monitor the regional oxygen gradients in the bone, and more accurate models are needed to unravel this topic [104]."
line 454: "One of the main challenge is to maintain an adequate level of oxygen within the tissue explant. Fluctuations in its concentration result in modified cell features; on the other hand, oxygen variations are typical of each tissue. In particular oxygen levels are very low in bone tissue in comparison to other sites. Moreover, commonly in vitro culture conditions are maintained under a determinate amount of oxygen (usually around 21%) [141], which is not representative of most tissues and especially not for bone. "
line 577:"Molecular oxygen is one of the most important variables to consider. In fact, oxygen transport is limited in vitro either by culture medium or by tissue thickness, leading to hypoxic areas, therefore restricting the viability of the explant. Several strategies can be followed to overcome these problems, like cultivate the samples into a bioreactor, using specific scaffolds resembling the microvasculature or oxygen carriers [102]. "